# Biallelic Loss-of-Function Variant in *MINPP1* Causes Pontocerebellar Hypoplasia with Characteristic Severe Neurodevelopmental Disorder

**DOI:** 10.3390/ijms26115213

**Published:** 2025-05-29

**Authors:** Aljazi Al-Maraghi, Rulan Shaath, Katherine Ford, Waleed Aamer, Jehan AlRayahi, Sura Hussein, Elbay Aliyev, Nourhen Agrebi, Muhammad Kohailan, Satanay Z. Hubrack, Sasirekha Palaniswamy, Adam D. Kennedy, Karen L. DeBalsi, Sarah H. Elsea, Ruba Benini, Tawfeg Ben-Omran, Bernice Lo, Ammira S. A. Akil, Khalid A. Fakhro

**Affiliations:** 1Laboratory of Genomic Medicine, Research Section, Sidra Medicine, Doha P.O. Box 26999, Qatar; aalmaraghi@sidra.org (A.A.-M.); rshath@sidra.org (R.S.); waamer@sidra.org (W.A.);; 2College of Health and Life Sciences, Hamad Bin Khalifa University, Doha P.O. Box 34110, Qatar; 3Laboratory of Immunoregulation, Research Section, Sidra Medicine, Doha P.O. Box 26999, Qatarshubrack@sidra.org (S.Z.H.); 4Department of Pediatric Radiology, Sidra Medicine, Doha P.O. Box 26999, Qatar; 5Metabolon Inc., Morrisville, NC 27560, USA; 6Department of Molecular and Human Genetics, Baylor College of Medicine, Houston, TX 77030, USA; sarah.elsea@bcm.edu; 7Neurology Division, Sidra Medicine, Doha P.O. Box 26999, Qatar; rbenini@sidra.org; 8Genetic and Genomic Medicine Division, Sidra Medicine, Doha P.O. Box 26999, Qatar; 9Department of Genetic Medicine, Weill Cornell Medical College, Doha P.O. Box 24144, Qatar

**Keywords:** pontocerebellar hypoplasia, neurodegenerative disorder, Middle East, rare disease, whole genome sequencing

## Abstract

Pontocerebellar hypoplasia (PCH) encompasses a group of autosomal recessive neurodegenerative disorders marked by cerebellar and pontine atrophy. Multiple subtypes of PCH have been identified, among which the rare subtype PCH type 16 is caused by *MINPP1* genetic variants. *MINPPI* encodes an enzyme essential for inositol polyphosphate dephosphorylation, regulating calcium and iron homeostasis. We conducted genome sequencing on a proband from the consanguineous family, who presented with a severe neurodegenerative disorder, to identify the underlying cause of disease. A comprehensive clinical assessment in addition to neuroradiological findings are described. We performed the functional validation of the identified variant and conducted untargeted metabolomic analyses. The clinical and radiological assessment of the patient showed a congenital brain anomaly and neurodegenerative symptoms. Further genetic analysis identified a homozygous loss-of-function variant (c.1401del, p.Ser468Valfs10*) in *MINPP1,* providing molecular confirmation of a clinical PCH diagnosis. While real-time quantitative PCR (RT-qPCR) showed that *MINPP1* gene expression was unaffected in the proband, Western blot analysis demonstrated reduced protein abundance, supporting a pathogenic role of the variant. Metabolomic profiling revealed elevated lipid levels and disrupted inositol metabolism, providing further insights into the disease mechanism. These findings establish the pathogenicity of the p.Ser468Valfs10* variant in *MINPP1* and highlight inositol metabolism as a potential pathway involved in PCH16, advancing the understanding of the pathophysiology of the disease.

## 1. Introduction

Pontocerebellar hypoplasia (PCH) is a rare group of neurodegenerative disorders characterized by the underdevelopment of the pons and cerebellum, which are critical structures in the brain responsible for coordinating movement, balance, and various other functions [1]. The disease is early onset, progressive in nature, and the precise incidence is currently unknown. In OMIM “https://www.omim.org/ (accessed 15 September 2024)”, PCH has been categorized into 17 distinct subgroups. These classifications are primarily determined by the underlying genetic factors, leading to distinct clinical, radiological, or biochemical characteristics. The two most common forms are type 1 (PCH1) and type 2 (PCH2), each presenting with distinct pathological features [2].

On a molecular mechanism level, PCH subtypes are usually characterized by defects in mRNA degradation, tRNA splicing, tRNA synthetase, and GTP synthesis [3]. Nonetheless, reports on other PCH subtypes revealed the involvement of genes in diverse pathways, including protein recycling (*VPS51*) [4], purine nucleotide cycle (*AMPD2*) [5], and inositol phosphate metabolism (*MINPP1*) [6,7]. Multiple Inositol Polyphosphate Phosphatase 1 (*MINPP1*) (MIM# 605391) is an enzyme that regulates intracellular inositol phosphate levels. The essential role of inositol phosphatases in central nervous system development has been demonstrated in mouse knockout experiments, which showed neural tube defects [8]. MINPP1 specifically converts hexakisphosphate (IP6) to inositol trisphosphate (IP3) intracellularly. A defective MINPP1 enzyme will result in an imbalance of the inositol polyphosphate metabolism through the increased chelation of related cations like iron and calcium [6].

Genetic variants in *MINPP1* are associated with PCH16 (MIM# 619527). So far, 11 variants in *MINPP1* have been reported in 16 patients. Affected subjects usually present with severe developmental delay, microcephaly, epilepsy, vision defects, ataxia, and muscular hyper and hypotonia [6,7].

Herein, we report a case involving a patient born into a consanguineous family where a homozygous loss-of-function (LOF) variant in *MINPP1* supported the diagnosis of PCH16. Integrating genomic analysis with untargeted metabolomics suggested an altered inositol pathway, highlighting multi-omic approaches in variant prioritization and diagnoses of rare mendelian disorders.

## 2. Results

### 2.1. Clinical Description

The index case is a 6-year-old Middle Eastern boy born to second-degree relatives. The mother has a history of two miscarriages, and three cousins had cerebellar malformation and epilepsy, with two dying in childhood. He was born at term via uncomplicated vaginal delivery but admitted to the NICU for transient tachypnea and treated with IV antibiotics for suspected sepsis.

In the first few months of life, he was found to have severe global developmental delays and subsequently never attained any developmental milestones. By the age of 3 months, he began to experience multiple generalized seizure types, most frequently myoclonic, brief tonic, and generalized tonic–clonic. Seizure patterns were recorded, consistent with Lennox–Gastaut syndrome. His seizures occurred daily and were refractory to multiple anti-seizure medications. By the age of 6 years, he was bed-bound, with no head support, and without the ability to roll, sit, or reach for objects. He also did not exhibit verbal communication or responsive smiles.

In addition to his severe neurodevelopmental disabilities, the patient had microcephaly (−3SD), recurrent respiratory infections, severe laryngomalacia, dysphagia, mild thoracolumbar scoliosis, and visual impairment. Motor examination revealed significant axial hypotonia with diffuse decreased muscle bulk and mild spasticity in the upper and lower limbs.

### 2.2. Neuroradiological Findings

Magnetic resonance imaging (MRI) of the brain was performed at the age of 4 months and repeated at the age of 2 years, showing significant, similar findings: marked atrophy and T2/FLAIR signal changes in the deep gray matter, primarily the basal ganglia and to a lesser extent, the ventrolateral thalamus; diffuse supratentorial volume loss and ex vacuo dilation of the ventricles; periventricular leukomalacia with enlarged ventricles showing a wavy outline; mild thinning of the corpus callosum; the figure “8” appearance of the lower mesencephalon; brainstem hypoplasia with markedly hypoplastic flattened pons with T2 signal changes involving the ventral pons; moderate-to-severe cerebellar hypoplasia, primarily involving the cerebellar hemisphere with relative preservation of the vermis, giving the “dragonfly sign”; cerebellar T2/FLAIR white matter hyper intensity. Moreover, the initial MRI at 4 months showed evidence of delayed myelination, which then restored to within normal range by 2 years of age (Figure 1).

### 2.3. Molecular Findings

The family history of parental consanguinity and the severity of symptoms prompted enrolling the family in the Mendelian Program at Sidra Medicine (Appendix A). Genome sequencing was performed on all family members and analyzed as described in “Methods”. A rare, homozygous variant (NM_004897.5: c.1401del, p.Ser468Valfs*10) in exon 5 of *MINPP1* was identified. Most in silico prediction tools do not provide a score for this variant, but the variant is well conserved (GERP score of 3.5). Validation by Sanger sequencing demonstrated that the patient was homozygous for the variant, while both parents and the sibling were heterozygous carriers (Appendix A). The variant was determined to be pathogenic according to ACMG criteria (PVS1, PS3, PS4, PM2, PP5). A summary of previously reported *MINPP1* variants is shown in Table 1.

RT-qPCR showed that there is no significant difference in the transcription of *MINPP1* between the patient and unrelated healthy donor controls (Figure 2B); however, Western blotting indicated that the patient had less abundant MINPP1 proteins than the healthy donor controls (Figure 2C), suggesting that the truncated protein is unstable and/or degraded.

### 2.4. Untargeted Metabolomics

A metabolomics analysis of plasma detected 743 molecules, including 680 molecules z-scored in comparison to the control population (Appendix A). This analysis revealed 98 clinically relevant biochemicals with levels below the 2.5 (n = 64) or above the 97.5 (n = 34) percentiles compared to a control reference population. Specifically, lipids formed >73% of the altered molecules, highlighting the role of *MINPP1* in lipid metabolism. As a group, phospholipids were the most dysregulated, representing >46% of all significantly altered molecules, including reduced levels of multiple plasmalogens and some sphingomyelins, with elevated levels of many phosphatidylcholines. The significant dysregulation of fatty acid metabolism was also evident, representing >28% of all altered molecules, with many medium- and long-chain saturated, long-chain mono- and polyunsaturated fatty acids, and dicarboxylic fatty acids significantly reduced (Figure 2D).

## 3. Discussion

In this study, we identified a pathogenic variant in *MINPP1* in a 6-year-old male patient born with a severe neurodegenerative disorder. The variant and clinical associations are consistent with previous reports of *MINPP1*-related PCH [6,7]. Radiologically, various degrees of pontine and cerebellar hypoplasia, global brain atrophy with or without corpus callosum thinning, basal ganglia atrophy, and signal changes are seen in most patients with PCH16 and, less commonly, thalamic hypoplasia or atrophy [6,7]. To our knowledge, atrophy of the basal ganglia and thalami is only described in PCH9 and PCH16 [9]; however, the figure “8” appearance of the midbrain is only found in our case and PCH9 [6,7,9] (Figure 1K).

The identified variant in *MINPP1* (c.1401del, p.Ser468Valfs*10) in the patient is located on the fifth exon (total size: five exons), in the ER retention peptide of the protein, and is expected to result in protein truncation, as the last twenty amino acids are substituted with nine different amino acids of the exon. Our data support the impact of this variant in producing an unstable truncated product, since protein expression was reduced (Figure 2C).

*MINPP1* deficiency affects the intracellular metabolism of phosphorylated inositols, influencing the levels of myo-inositol and 3-phosphoglycerate [6]. This study explores the extracellular plasma effects and dysregulated pathways associated with *MINPP1* deficiency. In the patient’s metabolome, two phosphorylated inositols, specifically 1-stearoyl-2-dihomo-linolenoyl-GPI and 1-palmitoyl-GPI, were significantly elevated. Myo-inositol and 3-phosphoglycerate, although present at a lower level, fell within the normal range, possibly due to the intracellular effects of *MINPP1* deficiency on these metabolites. Several phosphatidylcholines (PCs) and lysophospholipids, along with other lipids, displayed significant elevations, which could indicate neurological dysfunction as lipid dysregulation has been implicated in neurological diseases, including neurodegenerative diseases [10]. Furthermore, the lipid dysregulation suggests potential alterations in inositol metabolism [11].

Previously, Ucuncu et al. demonstrated that loss-of-function variants in *MINPP1* could disrupt cytosolic cation homeostasis, including calcium and iron. Consistent with this, the patient’s metabolomic profile showed significantly reduced levels of biliverdin and bilirubin, key byproducts of heme breakdown and involved in iron homeostasis signaling [12]. Clinically, the patient’s low hemoglobin levels (99 g/L) supported an impact on iron homeostasis. Moreover, sulfur/sulfo-containing molecules are elevated, which could reflect mitochondrial iron–sulfur dysfunction previously associated with neurodegenerative disorders [13]. These findings support disruption in iron homeostasis, potentially due to the loss-of-function of *MINPP1*.

Mitochondrial dysfunction was reported with PCH previously [3]. There is evidence that MINPP1 may serve as an intermediary between ER stress and mitochondrial apoptosis [14]. In addition to the mitochondrial iron–sulfur dysfunction, the patient’s metabolomic profile revealed low levels of plasmalogens and sphingomyelins, which are significant in peroxisomal and mitochondrial dysfunction [15]. The metabolic findings from this study support a role of MINPP1 in the mitochondria.

## 4. Materials and Methods

### 4.1. Ethical Approval

The study protocol received ethical approval from the Institutional Review Board of Sidra Medicine (IRB 1636872). Written informed consent was obtained for all participants.

### 4.2. Genome Sequencing and Variant Prioritization

DNA extraction was performed using the DNeasy Blood & Tissue Kit (Qiagen Sciences LLC, Germantown, MD, USA). Libraries for genome sequencing were prepared through the TruSeq DNA Nano kit (Illumina, San Diego, CA, USA). The samples from the parents, affected child, and healthy sibling were sequenced using the Illumina platform, to an average depth of 30×. Data annotation was performed using an established in-house pipeline. All variants were mainly prioritized based on a mean allele frequency (MAF) of less than 1% in databases such as the Genome Aggregation Database (GnomAD) [16] and Qatar Genome Project Database [17], the mode of inheritance, and in silico prediction tools. Analysis yielded eight rare homozygous variants, including a predicted loss-of-function (LoF) variant in *MINPP1* being the only gene clinically relevant to the patient’s phenotype. Sanger sequencing was performed using the following *MINPP1* primer sequence: Forward: 5′-ACAATTCCGAGTGCAGATGT-3′; Reverse: 5′-GGAATTGCCTACCTATTACAAGC-3′.

### 4.3. Untargeted Metabolomics Sample Preparation and Analysis

For the index case, Baylor Genetics and Metabolon, Inc., carried out clinical-grade untargeted metabolomics using ultrahigh-performance liquid chromatography–tandem mass spectroscopy (UPLC-MS/MS) analysis, as previously described [18,19]. Briefly, peripheral blood was collected in EDTA-coated tubes, and 500 µL of residual plasma was extracted and shipped overnight in frozen condition (−80 °C) on dry ice to Baylor Genetics. The sample was analyzed using the UPLC-MS/MS platform, which enables the broad detection of compounds with diverse physicochemical properties by incorporating different chromatography techniques and mass spectrometry ionization modes. We compared clinically relevant biochemicals with levels above the 97.5 or below the 2.5 percentiles to an established reference population consisting of 399 presumably healthy individuals analyzed at the Baylor Clinical Biochemical Genetics Laboratory. This pooled control group included 164 females (age range: 0.5 to <19 years; mean age: 5.9 years) and 235 males (age range: 0.04 to <19 years; mean age: 6.1 years) from diverse ethnicities [20].

### 4.4. Cell Culture

Peripheral blood mononuclear cells (PBMCs) from patients and healthy donors were isolated from whole blood by Ficoll gradient separation. Activated T cells (ATCs) were expanded from PBMCs in complete RPMI 1640 media with 10% fetal calf serum (FCS) by stimulating with 1 mg/mL of antibodies against CD3 (clone HIT3a, BD Pharmingen, Franklin Lakes, NJ, USA) and CD28 (clone CD28.2, BD Pharmingen) for 3 days, then further grown in the presence of 100 U/mL of IL-2 for 5–10 days. Epstein–Barr virus-immortalized B cells (EBV-B) were generated by infecting PBMCs with EBV from B95-8 culture supernatant in RPMI 1640 media supplemented with 10% FCS and cyclosporin A. Starting 1 week post-infection, fresh complete RPMI 1640 was added periodically for approximately a month to generate a stable EBV-B cell line.

### 4.5. Gene Expression and Western Blot Analysis

Total RNA was isolated from the patient’s ATCs and EBV-B cells using the RNeasy Mini Kit (Qiagen, #74104) and was purified from DNA contamination using DNase 1 (Invitrogen, #18068015). Reverse transcription was performed by the SuperScript IV First-Strand Synthesis System (Invitrogen, Waltham, MA, USA, #18091050), and transcripts were amplified by real-time quantitative PCR using Fast SYBR Green Master Mix (Applied Biosystems, Waltham, MA, USA, #4385612). Primer sequences were as follows: for RPLP0, 5′-TGGTCATCCAGCAGGTGTTCGA-3′ and 5′-ACAGACACTGGCAACATTGCGG-3′; for GAPDH, 5′-CTGACTTCAACAGCGACACC-3′ and 5′-TTACTCCTTGGAGGCCATGT-3′; for *MINPP1*, 5′-GCACTTGGACAAAGCAGTTGA-3′ and 5′-GGCATAAGGTACAATGAGACCAC-3′. The resulting comparative cycle threshold (ΔΔCT) data were used to calculate the mean fold change (FC) in expression between *MINPP1* and the reference genes, normalized to the healthy donors. Error bars reflect the standard error, and statistical significance was calculated by two-tailed, heteroscedastic Student’s *t*-Test (*p* < 0.05).

The patient’s activated T-cells and EBV-immortalized B cells were grown in 10% RPMI, lysed, and sonicated in RIPA buffer containing HALT protease inhibitor. Lysates were quantified, normalized, and loaded into a 10% gel (20 ug/well) for SDS-PAGE followed by semi-dry transfer onto the PVDF membrane. Membranes were blocked in milk, incubated with mouse anti-MIPP-A8 monoclonal antibody (sc-514214, Santa Cruz Biotechnology, Dallas, TX, USA) in 3% BSA overnight at 4 °C, then with HRP-conjugated goat anti-mouse IgG (#5450-0011, SeraCare, Milford, MA, USA) in 5% milk for one hour at room temperature, and visualized with SuperSignal West Femto (Thermofisher Scientific, 34096, Waltham, MA, USA). Membranes were stripped using Restore WB Stripping Buffer (#21059, Thermo Scientific) before incubation in HRP-conjugated mouse anti-beta-actin antibody (ProteinTech, Rosemont, IL, USA, HRP-60008) in 5% milk for 30 min at room temperature and visualized with SuperSignal West Femto. Control ATCs and EBV-B cells were sourced from healthy adult donors of both genders.

## 5. Conclusions

In conclusion, this is the first report to provide functional evidence supporting the pathogenicity of the *MINPP1* (c.1401del) variant in a 6-year-old male. Our findings also shed light on the potential of metabolomics in providing valuable insights on disease pathophysiology and identifying novel pathway alterations. While PCH treatment is primarily symptomatic, accurate molecular diagnosis is crucial for informing parents regarding disease prognosis and offering tailored reproductive planning for future pregnancies. This is particularly important in the setting of consanguineous families to mitigate the risk of recurrence in future pregnancies.

## Figures and Tables

**Figure 1 ijms-26-05213-f001:**
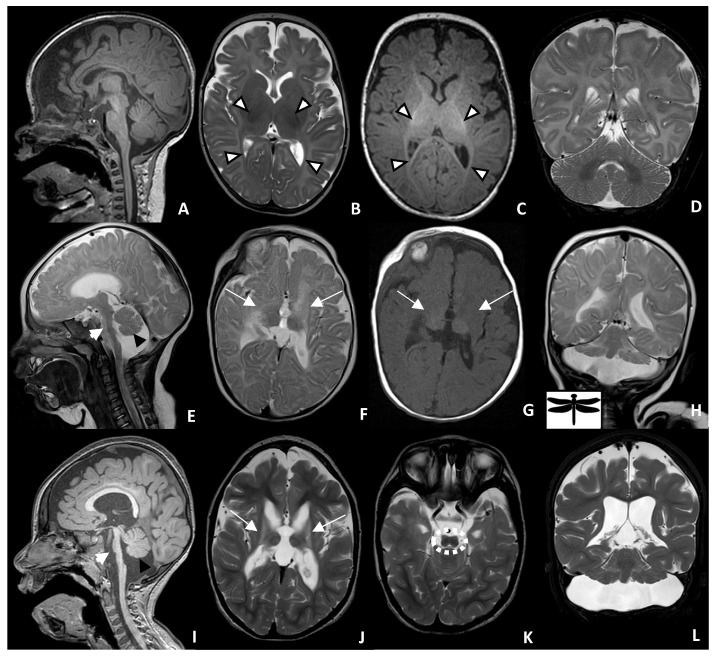
Brain MRI of a 4-month-old control (**A**–**D**) and patient with PCH-16 at 4 months of age (**E**–**H**) and at 2 years of age (**I**–**L**). Sagittal T1 of the control (**A**) shows normal brainstem and cerebellar size and configuration. Axial T2 (**B**) and axial T1 (**C**) of the brain show normal, dark T2 and a bright T1 myelin signal in the internal capsule and optic radiation (white arrowhead). Sagittal T2 (**E**) and T1 (**I**) of the patient reveal brainstem atrophy, with markedly hypoplastic and flattened pons (short white arrow), as well as a small cerebellar vermis (black arrowheads). Axial T2 (**F**) and T1 (**G**) of the patient at 4 months of age show an absent myelin signal within the internal capsule and optic radiation, as well as periventricular leukomalacia and atrophic basal ganglia and thalamus (long white arrows). This can be better seen on the axial T2 (**J**) image, taken at 2 years of age. Coronal T2 (**H**,**L**) demonstrates a hypoplastic cerebellum with greater involvement of the cerebellar hemisphere relative to the vermis, giving the “dragonfly” appearance. Axial T2 (**K**) demonstrates the figure “8” appearance of the lower midbrain (dashed white circle).

**Figure 2 ijms-26-05213-f002:**
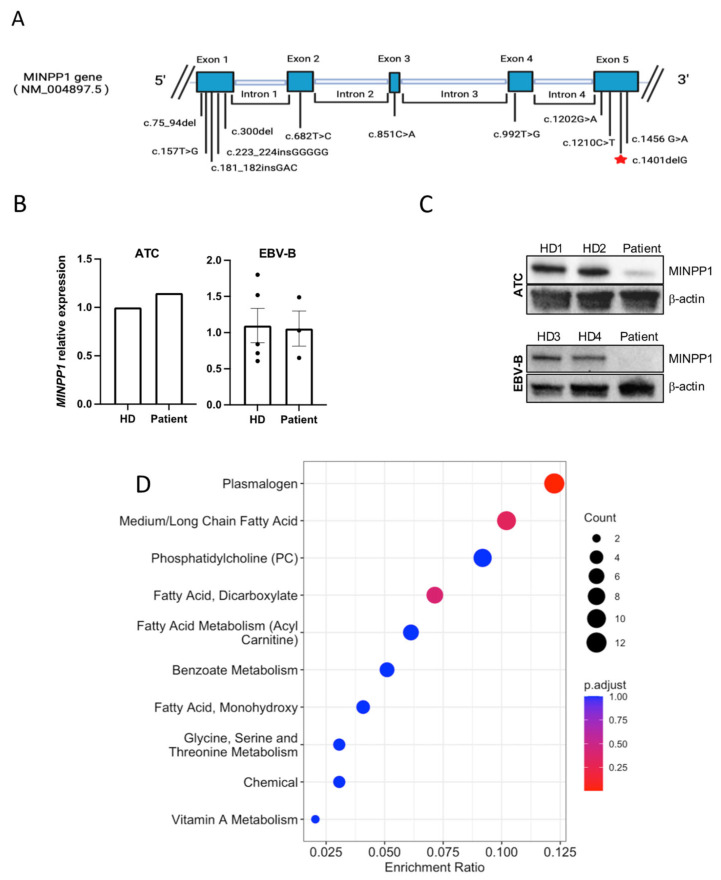
(**A**) Schematic of the *MINPP1* gene with highlights of all known variants; red star represents the variant reported in our patient. (**B**) RT-qPCR results showing *MINPP1* relative expression levels from patient versus healthy donor (HD) activated T cells (ATCs) or EBV-B cell lines. There was no significant difference in *MINPP1* transcript levels between the patient and controls. The housekeeping gene used was *GAPDH* for ATCs and *RPLP0* for EBV-B. For ATCs, the data shown are the mean of two technical duplicates, normalized to a single healthy donor (patient’s FC = 1.147; HD’s FC = 1.00). For EBV-B, the data shown are the mean +/− standard error of HD3 and HD4, each with 2–3 replicates (patient’s FC = 1.06 +/− 0.241; combined HD average FC = 1.10 +/− 0.237; *p* = 0.13). (**C**) Western blot results show that the truncated MINPP1 protein is less abundant in the patient than in healthy donors (HD1–4). (**D**) Pathway enrichment analysis including metabolites with z-scores ≤ 2 and ≥2. The size of the point represents the number of different metabolites enriched in the corresponding pathway (count), while the redder the enrichment, the closer the *p* value is to 0.

**Table 1 ijms-26-05213-t001:** Summary of reported *MINPP1* variants.

	Number of Affected	Position (GRCh37\hg19)	HGVS cDNA	HGVS Protein	Exon	Zygosity
Present report	1	chr10:89312171	c.1401del	p.Ser468Valfs*10	5	Homozygous
Appelhof et al. [7]	2	chr10:89264735	c.75_94del	p.Leu27Argfs*39	1	Homozygous
	2	chr10:89272896	c.851C>A	p.Ala284Asp	3	Homozygous
	1	chr10:89311981	c.1210C>T	p.(Arg404*)	5	Homozygous
	3	chr10:89280851	c.992T>G	p.(Ile331Ser)	4	Homozygous
Ucuncu et al. [6]	2	chr10:89264893	c.223_224insGGGGG	p.Glu75Glyfs*30	1	Homozygous
	1	chr10:89264829 chr10:89264971	c.157T>G c.300del	p.Tyr53Asp p.Lys101Serfs*2	1 1	Compound Heterozygous
	1	chr10:89311973	c.1202G>A	p.Arg401Gln	5	Homozygous
	1	chr10:89312227	c.1456G>A	p.Glu486Lys	5	Homozygous
	1	chr10:89264853	c.181_182insGAC	p.Leu61*	1	Homozygous
	2	chr10:89268137	c.682T>C	p.Phe228Leu	2	Homozygous

## Data Availability

The datasets analyzed during the current study are available in the Genome Sequence Archive in Sidra Medicine, Qatar.

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
