# Peer review of "Biallelic Loss-of-Function Variant in MINPP1 Causes Pontocerebellar Hypoplasia with Characteristic Severe Neurodevelopmental Disorder"

_ijms, 2025, doi:10.3390/ijms26115213_

Round 1
Reviewer 1 Report
Comments and Suggestions for Authors
Although the manuscript by Al-Maraghi et al. shows interesting results, some major issues prevent its publication in its current form.
It would be useful to describe other mutations related to PCH16, their position in the protein and compare them with this new mutation located in the ER retention peptide. The manuscript by Appelhorf et al. published in EJHG in 2020 may be useful for this purpose.
According to the instructions for authors (https://www.mdpi.com/journal/ijms/instructions), my suggestion is to reclassify this article as a "case report" due to the nature of the work. The introduction is short for a research article, but probably appropriate for a case report. In addition, case reports should include "a conclusion briefly outlining the take-home message and the lessons learned", that is missing in the manuscript (probably the last paragraph could be used in this way)
Since protein levels but not RNA levels are affected, the proposed mechanism is an effect on protein stability, which could be demonstrated using proteosome inhibitors such as MG132. In addition, the mutation is localised at the end of the protein, in the ER retention domain, so stability impairment could be an explanation. Other disease-related mutations may also be explained in terms of phenotype, protein domain and possible effect on structure, stability or function.
RT-PCR should be called RT-qPCR and abbreviations should be defined. In addition, statistics and numbers should be presented.
Figure 2 should be enlarged. It is impossible to see properly at this size. Consider separate and develop panel F in another figure, more explanatory. Metabolic analysis methods and results should be better explained and discussed.
In silico and qPCR analyses should be adequately described.
Minor issues
Gene names should be written in italics.
Abstract. MINPP1 is used for molecular diagnosis, but the main diagnosis should be clinical.
Author Response
Please see PDF attachment for a point-by-point response to the reviewer's comments.

Reviewer 2 Report
Comments and Suggestions for Authors
This manuscript presents a well-documented case study of pontocerebellar hypoplasia type 16 (PCH16) caused by a biallelic loss-of-function variant in MINPP1. The integration of whole genome sequencing, radiological imaging, protein expression analysis, and untargeted metabolomics is commendable and reflects a comprehensive, multi-omic approach. The manuscript is clearly written, and the clinical phenotype is thoroughly described.
However, a major issue that must be addressed before publication is the lack of clarity regarding the source and characteristics of the control samples used in the protein expression and metabolomic analyses. It is not stated whether these controls were matched to the patient by age and sex. Given the significant developmental, metabolic, and physiological changes that occur across pediatric age ranges—and the known sex differences in metabolism—matching to appropriately selected controls is essential for valid interpretation of differences. Without this information, the extent to which the observed findings (e.g., altered lipid metabolism, reduced protein abundance) are attributable to the MINPP1 variant rather than developmental or demographic variability remains uncertain. The manuscript should explicitly state the characteristics of the control cohort and justify the use of these samples in comparison to the proband.
In addition, while the case adds value to the growing literature on PCH16, the authors should more clearly define the novel aspects of this particular case—especially in light of several prior reports on MINPP1-related disease. This would help contextualize the significance of the findings and their contribution to the field.
I recommend major revision. Addressing the concerns about control matching is essential, and strengthening the case’s novelty and relevance would further improve the manuscript.
Author Response
Please see the attachment for a point-by-point response to the reviewer's comments

Round 2
Reviewer 1 Report
Comments and Suggestions for Authors
The revised version of the manuscript is not yet ready for publication.
The in silico methods used are still not well described.
The original figure in the Western blot is not correct, as it is the same as in the manuscript.
The electrophoregrams shown, previously in the figure in the main text, now in the supplementary material, do not correspond to the family shown in the pedigree. The "healthy donor" panel should be the electrophoregram of a heterozygote of the panel. The result should therefore be an electrophoregram with superimposed peaks from the deletion position. Furthermore, in the homozygote for the deletion, the 2 A peaks are strange, so they don't completely disappear. They should be like the other 3 adenines at the end of the electrophoregram figure.
Author Response
Please see attachment for point-by-point response to reviewer's comments

Reviewer 2 Report
Comments and Suggestions for Authors
The authors have answered all of my comments.
Author Response
Thank you for the feedback, no further comments were provided.
Round 3
Reviewer 1 Report
Comments and Suggestions for Authors
Although the reviewer's concerns are well addressed, I can't see the answers in the manuscript or the supplementary material (e.g. the heterozygote electropherogram in Supplementary Figure 1B). Furthermore, if in silico methods do not provide results, this should be indicated in the text.
Author Response
Dear reviewer, thank you for the comments, please see the attachment.
